biomechanics; cell wall loosening; cell wall remodelling; development; expansin; plant.

**Author for correspondence:**
J. Hejatko,
E-mail: jan.hejatko@ceitec.muni.cz

# Expansin-mediated developmental and adaptive responses: A matter of cell wall biomechanics?

Marketa Samalova[1,2] , Evelina Gahurova[1,3] and Jan Hejatko[1,3]

[1]CEITEC - Central European Institute of Technology, Masaryk University, Brno, Czech Republic; [2]Department of Experimental Biology, Faculty of Science, Masaryk University, Brno, Czech Republic; [3]National Centre for Biotechnological Research, Faculty of Science, Masaryk University, Brno, Czech Republic

## Abstract

Biomechanical properties of the cell wall (CW) are important for many developmental and adaptive responses in plants. Expansins were shown to mediate pH-dependent CW enlargement via a process called CW loosening. Here, we provide a brief overview of expansin occurrence in plant and non-plant species, their structure and mode of action including the role of hormone-regulated CW acidification in the control of expansin activity. We depict the historical as well as recent CW models, discuss the role of expansins in the CW biomechanics and address the developmental importance of expansin-regulated CW loosening in cell elongation and new primordia formation. We summarise the data published so far on the role of expansins in the abiotic stress response as well as the rather scarce evidence and hypotheses on the possible mechanisms underlying expansin-mediated abiotic stress resistance. Finally, we wrap it up by highlighting possible future directions in expansin research.

## 1. Introduction

The primary plant cell wall (CW) is a multi-layered structure in which each layer (lamella) consists of load bearing cellulose microfibrils laterally interconnected possibly with xyloglucan and embedded into a pectin matrix (Zhang et al., 2019a; 2021a). The properties of CW are being constantly modified to allow for morphological changes that are necessary for plant growth and development both in the shoot (Gruel et al., 2016; Hamant et al., 2008; Hervieux et al., 2017; Landrein et al., 2015; Majda et al., 2017; Pien et al., 2001; Reinhardt et al., 1998; Sampathkumar et al., 2014; Takatani et al., 2020) and root (Barbez et al., 2017; Hurny et al., 2020; Mielke et al., 2021; Pacifici et al., 2018; Ramakrishna et al., 2019; Vermeer et al., 2014). Mechanical properties of the CW are regulated by a variety of agents including expansins (Cosgrove, 2000; McQueen-Mason et al., 1992), glucanases (Yoshida & Komae, 2006; Yuan et al., 2001; Zhang et al., 2019a), pectin methylesterases (Goldberg et al., 1996; Peaucelle et al., 2008; Wang et al., 2020), calcium ions (Bou Dahner et al., 2018; Wang et al., 2020) and others. While endoglucanases and other enzymes typically decrease the number of linkages between cellulose and other CW molecules (i.e., mediate CW remodelling, see the Glossary) leading to a weaker (i.e., more easily breakable) wall, α-expansins induce creep—an irreversible time-dependent CW enlargement (Cosgrove, 2016a; Park & Cosgrove, 2012a; Wang et al., 2013; Yuan et al., 2001). These types of biomechanical modifications should be distinguished. Thus, the timing and location of growth are controlled by spatial- and time-specific modification of the mechanical properties of the CW. Here we review recent contributions on the role of α-expansins in the control of biomechanical CW properties, focusing primarily on their role in plant development and abiotic stress response.

## 2. Expansin discovery and evolution

Expansins were discovered in plants as proteins that play a crucial role in CW loosening (McQueen-Mason et al., 1992), as they induce stress relaxation and extension in plant CWs during pH-dependent 'acid growth' (Rayle & Cleland, 1992). Since then, expansins have been

shown to be involved in many aspects of plant growth and development. Expansins are present to the best of our knowledge in all plant species, although some gene loss is observable in highly adapted aquatic species (Hepler et al., 2020). Expansins can also be found in fungi and bacteria, probably as a result of horizontal gene transfer (Georgelis et al., 2015). However, the presence of these genes in all eukaryotic microorganisms that use cellulose as a structural component of their CW suggests that expansins evolved in ancient marine microorganisms long before the evolution of land plants (Chase et al., 2020). Expansins from diverse bacteria and fungi assisting plant–microbe interactions in nature have often been utilised in industrial applications to facilitate lignocellulose degradation that is used further in the conversion of biomass into alternative fuels (Georgelis et al., 2015; Liu et al., 2015).

## 3. The expansin (super)family

Based on phylogenetic sequence homology, four distinct genetic subfamilies of expansins are currently recognised in vascular plants: $\alpha$-expansin (EXPA), $\beta$-expansin (EXPB), expansin-like A (EXLA) and expansin-like B (EXLB) (Sampedro & Cosgrove, 2005). Two of these subfamilies, the $\alpha$ and $\beta$ expansins have been demonstrated experimentally to induce CW loosening (Cosgrove et al., 1997; McQueen-Mason et al., 1992). EXPA is the most numerous subfamily, for example in *Arabidopsis thaliana* there are 26 *EXPA* genes, 6 *EXPB*, 3 *EXLA* and 1 *EXLB*. Apart from *Arabidopsis*, rice and poplar (Sampedro & Cosgrove, 2005), genome-wide identification and expression profile analysis of expansin gene families have recently been performed in sugarcane (Santiago et al., 2018), wheat (Han et al., 2019; Zhang et al., 2018a), potato (Chen et al., 2019), Chinese jujube (Hou et al., 2019), cotton (Lv et al., 2020) and *Brassica* species (Li et al., 2021a).

Although the main focus of this review is on EXPA, it is worth mentioning that the group of $\beta$-expansins expanded significantly in grasses (Sampedro et al., 2015). As an example, EXPB1 (also called Zea m 1) is a member of group-1 grass pollen allergens and its crystal structure has been resolved suggesting the role of EXPB1 in the local movement and stress relaxation of (arabino)xylan-cellulose networks within the wall (Yennawar et al., 2006). Detailed characterisation of EXPB1 function in extracted maize CWs revealed that the protein primarily binds glucuronoarabinoxylan, the major polysaccharide in grass CWs (Wang et al., 2016a) that is largely absent in primary CWs of dicots (Carpita, 1996; Vogel, 2008). In maize, the group is needed for pollen separation and stigma penetration (Valdivia et al., 2009).

## 4. Expansin structure and mode of action

### 4.1. Expansin structure

Expansins are modular, torpedo-shaped proteins that consist of two tightly packed, structured domains of 200–250 amino acids, connected by a short linker and preceded by a signal peptide. The *N*-terminal domain (D1) is a six-stranded double-psi ($\omega$) $\beta$-barrel related to family 45 glycoside hydrolases (GH45), but lacks the critical catalytic Asp required for hydrolytic activity (Cosgrove, 2015; Georgelis et al., 2015; Kerff et al., 2008; Yennawar et al., 2006). The *C*-terminal domain (D2) with a $\beta$-sandwich fold is related to group-2 grass pollen allergens and resembles the carbohydrate binding module (CBM) family 63 (Chase et al., 2020; Georgelis et al., 2012). Both domains are required for full CW loosening activity (Georgelis et al., 2011; Sampedro & Cosgrove, 2005). The

Expansin Engineering Database (ExED; https://exed.biocatnet.de) is a useful navigation and classification tool for expansins and their homologues and is based on newly created profile hidden Markov models of the two expansin domains (Lohoff et al., 2020).

Despite the rather long history of expansin research, many of the details of the functional and structural properties underlying the molecular mechanism of expansin action in enabling CW expansion still remain undiscovered. One of the reasons for this knowledge gap is that, unlike bacterial or fungal expansins, plant $\alpha$-expansins have proven difficult to produce in the active form using heterologous expression systems (Gaete-Eastman et al., 2015). Nonetheless, computational 3D models built through comparative modelling and molecular dynamics simulations have yielded the first structural approximation of several $\alpha$-expansins (Gaete-Eastman et al., 2015; Mateluna et al., 2017; Pastor et al., 2015; Valenzuela-Riffo et al., 2018; 2020) and confirmed that expansins can form a stable complex with cellulose via the flat aromatic surface of the *C*-terminal domain (Valenzuela-Riffo et al., 2018). Based on the model, the expansins also interacted with the xyloglucan XXFG ligand, but were less likely to bind the XXXG ligand; they did not interact with pectin (Valenzuela-Riffo et al., 2020), the latter being in contrast to experimental data (Nardi et al., 2013). Recently, the protein structure of several expansins was determined by the AlphaFold protein prediction algorithm (Figure 1a) proven to be highly reliable in terms of the predicted protein structure (Jumper et al., 2021; Varadi et al., 2021).

### 4.2. Bacterial expansins

Because of the aforementioned limitations, our knowledge of the mode of expansin action at atomic resolution is limited to bacterial expansins. Cellulose binding was demonstrated for *Bacillus subtilis* expansin EXLX1, a bacterial expansin that can loosen plant CWs. Through hydrophobic interactions of three linearly arranged, highly conserved aromatic residues (W125, W126 and Y157) in the D2 domain, EXLX1 binds tightly to crystalline cellulose rather than to linear oligosaccharides (Boraston et al., 2001; Georgelis et al., 2012; Kim et al., 2013). Molecular dynamics simulations suggest that the expansin has both a cellulose-weakening and a cellulose-binding activity that depends on substrate crystallinity (Orłowski et al., 2018). Indeed, adsorption of EXLX1 onto a cellulose film decreased the crystallinity index, disrupted hydrogen bonding, and increased the surface area of cellulose, indicating greater accessibility of the substrate to proteins (Duan et al., 2018). It is this characteristic that makes expansin and expansin-like proteins that act synergistically with cellulases during hydrolysis useful for industry, and they are often used as biological pre-treatments to disrupt and open up recalcitrant lignocellulose complexes for industrial applications (Georgelis et al., 2011; 2015; Kerff et al., 2008; Kim et al., 2009).

Other investigations of EXLX1 adsorption onto cellulose, using quartz crystal microbalance with dissipation (QCM-D), confirmed that cellobiose and xylose enhanced EXLX1 adsorption at low concentrations but inhibited it at high concentrations (Zhang et al., 2020). Monitoring real-time adsorption of endo/exo-glucanases with EXLX1 and the enzymatic hydrolysis of cellulose showed synergistic effects. This increased activity can be due to easier access of the cellulase to the cellulose chains, but other effects such as electrostatic or other physical interactions between the adsorbed EXLX1 and cellulases cannot be ruled out (Zhang et al., 2021b). However, bacterial expansins have much weaker cellulose binding and wall-loosening activity than plant $\alpha$-expansins (Kerff et al.,

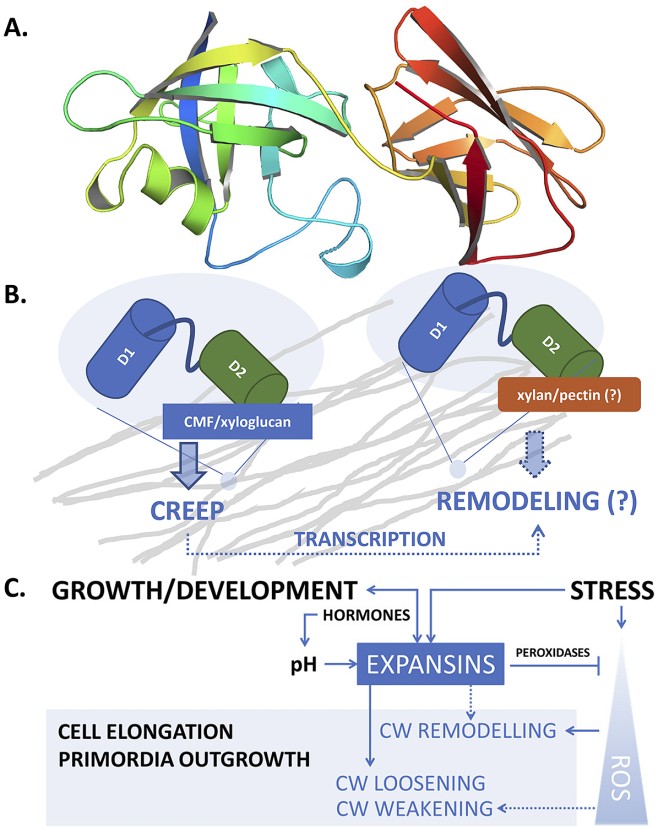

**Fig. 1.** (a) Structure of AtEXPA1 determined by the AlphaFold algorithm. *N*-terminal six-stranded double-psi ($\omega$) $\beta$-barrel D1 domain related to family 45 glycoside hydrolases (GH45) (green/blue, left) and *C*-terminal $\beta$-sandwich fold D2 domain related to group-2 grass pollen allergens resembling the carbohydrate binding module (CBM) family 63 (red/orange, right); the unstructured signal peptide is not shown. (b) Upon binding the load-bearing cellulose microfibril (CMF) network laterally interconnected with possible xyloglucan contribution (grey), expansins induce CW expansion via CW creep. By interfering with CW remodelling enzymes via binding to xylan and/or pectin or through transcriptional feedback regulations in a response to changed CW biomechanics, expansins might contribute to CW remodelling, too. (c) Expansin expression and localization is regulated during plant development, ensuring expansin action in a manner that is specific to their dose and the particular developmental context. Conversely, expansin action on CW biomechanics affects plant development and growth responses by regulating cell elongation and/or primordia specification/outgrowth. Expansins are activated in response to various stresses associated with ROS production. Expansin expression might be mediated by developmental- and stress-regulated hormone production, controlling expansin activity also via spatial-specific CW acidification. Expansins could mitigate ROS effects by upregulating CW peroxidases. In turn, ROS also contribute to the regulation of CW biomechanical properties. While short-term or low-level ROS production leads to growth inhibition by inducing crosslinking of CW components, high ROS levels/long-term ROS production leads to OH$^\circ$-radical formation that was hypothesised to allow restoration of cell expansion via polymer cleavage, leading to CW weakening. See the main text for a more detailed description.

2008; Kim et al., 2009), and recent results suggest that although EXLX1 is homologous with plant expansins, it possibly has distinct effects on plant CWs (Hepler & Cosgrove, 2019).

### 4.3. Expansin-mediated CW loosening

According to the loosening theory (Cosgrove, 2015), well-hydrated non-growing cells reach osmotic equilibrium with wall stresses counter-balancing the outward turgor pressure against the wall. In growing cells, however, walls are loosened (primarily via pH-dependent action of expansins), which means that the

load-bearing part of the wall is relaxed, releasing the tensile stress and simultaneously reducing cell turgor. Consequently, water flows into the cell, expanding the wall and restoring turgor and wall stress, together driving cell growth (Cosgrove, 2015; 2018a). Importantly, cell expansion starts with CW loosening/relaxation, followed by a decrease and a subsequent increase of cell turgor, not vice versa (Cosgrove, 1993).

There is a significant body of evidence suggesting that expansins themselves are incapable of hydrolysing the polysaccharide substrate itself (Kerff et al., 2008; McQueen-Mason & Cosgrove, 1995; McQueen-Mason et al., 1992). Nevertheless, pH-dependent, expansin-mediated CW loosening promotes relaxation of the CW structure, thus contributing to CW remodelling by allowing different hydrolases to access their polysaccharide substrates (Cosgrove, 2000; 2005; Whitney et al., 2000).

### 4.4. Apoplast acidification is necessary for expansin-mediated cell expansion

According to the 'acid growth theory' (Hager et al., 1971; Rayle & Cleland, 1970), auxin triggers extrusion of protons (H$^+$) into the apoplast, which activates expansins that subsequently loosen the CW and allow growth (McQueen-Mason et al., 1992). The most important players in this process are plasma membrane P-type H$^+$-ATPases which pump out protons to the wall matrix, consequently leading to apoplast acidification (Takahashi et al., 2012). Later it was discovered that the transport inhibitor response1/auxin signaling F-box—auxin/indole-3-acetic acid (TIR1/AFB-Aux/IAA) auxin signalling machinery transcriptionally upregulates the SMALL AUXIN UP-RNA 19 (SAUR19) expression levels (Fendrych et al., 2016). SAUR19 inhibits the activity of TYPE 2C PROTEIN PHOSPHATASES (PP2C), thus maintaining the H$^+$-ATPase in an active state (Spartz et al., 2014). Pumping protons causes plasma membrane hyperpolarisation and also activates K$^+$ channels that (in a short term) electrically balance the H$^+$ efflux and (in the long term) maintain intracellular osmotic potential low, thus allowing sustained water uptake and turgor pressure forcing the CW to extend (Thiel & Weise, 1999; for review see Arsuffi & Braybrook, 2018).

Given the different effects of auxins on shoots compared with roots (for review see Du et al., 2020; Dunser & Kleine-Vehn, 2015; Li et al., 2021b), the acid growth theory seems to be more complex in roots, suggesting possible non-transcriptional regulations (Pacheco-Villalobos et al., 2016). In line with that, the non-transcriptional branch of the cytosolic TIR1/AFB pathway was demonstrated to trigger a rapid Cyclic Nucleotide-Gated Channel 14 (CNGC14)-mediated Ca$^{2+}$ influx and an unknown channel or transporter-mediated H$^+$ influx leading to apoplast alkalization inhibiting the growth (Fendrych et al., 2018; Li et al., 2021b). Recently, it was shown that the cell surface-based TRANSMEMBRANE KINASE1 (TMK1) directly binds and activates plasma membrane H$^+$-ATPases thus promoting CW acidification in both shoots and roots (Li et al., 2021c; Lin et al., 2021), acting antagonistically to the noncanonical TIR1/AFB pathway (Li et al., 2021b).

However, not only auxin can control apoplastic pH. Cytokinins were proposed to upregulate the expression of genes for H$^+$-ATPases AHA2 and AHA7, facilitating thus EXPA1-mediated induction of cell elongation in the root transition zone (Pacifici et al., 2018). Furthermore, Großeholz et al. (2021) recently proposed a new model in which brassinosteroid-mediated cell elongation response depends on the amount and activity of

$H^+$-ATPases in the plasma membrane. Also here, the $K^+$ antiport, this time mediated via CNGC10, is necessary to compensate for $H^+$ efflux, thus keeping the plasma membrane potential constant. Using microelectrode ion flux estimation measurements, Großeholz et al. (2021) demonstrated net $H^+$ influx in the root meristematic zone while $H^+$ efflux in the root transition zone. The resulting pH gradient is proposed to be instructive for the cell elongation in the root transition/elongation zone. Altogether, not only the spatiotemporal specificity of *EXPAs* expression and protein localization but also the spatial-specific control over $H^+$ fluxes leading to the changes in the apoplastic pH are important factors controlling the EXPA-mediated cell expansion.

## 5. Expansins and CW biomechanics

### 5.1. Historical overview of the primary CW models

Previous depictions of accepted CW models (Carpita & Gibeaut, 1993; Fry, 1989; Hayashi, 1989; Nishitani, 1998) presented cellulose microfibrils as well-spaced and non-contacting rods with xyloglucan covering most cellulose surfaces and tethering them together to form the load-bearing network. Indeed, it was confirmed that enlargement of the CW required separation of cellulose microfibrils; however, high resolution (FESEM and AFM) images from slowly extended CWs in vitro and control non-extended samples, appeared indistinguishable (Marga et al., 2005). CW can therefore extend slowly through creep but without passive reorientation of the innermost microfibrils, suggesting that the loosening agents act selectively on the cross-linking polymers between parallel microfibrils, rather than more generally on the wall matrix, increasing microfibril spacing but without reorienting them (Marga et al., 2005).

In 2008, Cavalier et al. (2008) showed that *Arabidopsis* xyloglucan-deficient (*xylosyltransferase1/xylosyltransferase2*; *xxt1/ xxt2*) mutant plants were reduced in size, but otherwise seemed to develop normally. Nevertheless, stress/strain assays performed by Park and Cosgrove (2012b) showed that the *xxt1/xxt2* walls were more pliant than wild-type (WT) walls but less extensible in the creep and stress-relaxation processes mediated by $\alpha$-expansin, suggesting that xyloglucan plays a CW strengthening role. Similarly, loosening agents that act on xylans and pectins elicited greater extension in creep assays of the mutant xyloglucan-deficient CWs, demonstrating that these polymers take on a larger mechanical role in the absence of xyloglucan. The results also indicated that growth reduction in *xxt1/xxt2* plants is likely due to the absence of the native target for CW loosening by $\alpha$-expansins (Park & Cosgrove, 2012b).

Although xyloglucan has the ability to bind tightly to cellulose, NMR analyses of complex CWs showed that very little of the cellulose microfibril surface is actually in contact with xyloglucan (Bootten et al., 2004; Dick-Perez et al., 2011). On the other hand, pectin content is approximately 3-fold that of xyloglucan in *Arabidopsis* CWs (White et al., 2014) and makes the majority of matrix contacts with cellulose surfaces. The binding of xyloglucan is restricted to a minor component that appears to be closely intertwined with cellulose at discrete sites designated as 'biomechanical hotspots' (Cosgrove, 2014; Park & Cosgrove, 2015). Indeed, substantial wall loosening by substrate-specific endoglucanases (CXEG) was traced to the digestion of a specific component comprising <1% of the xyloglucan in the wall, indicating that only a small number of sites may control wall extensibility (Park & Cosgrove, 2012b). This picture of a few biomechanical junctions is also consistent with

the low density of $\alpha$-expansin binding sites in the CW (McQueen-Mason & Cosgrove, 1995).

The biomechanical '*hotspot hypothesis*' proposes that wall extensibility is controlled at discrete sites where microfibrils come into close contact with one another (Zhang et al., 2014) via a monolayer of xyloglucan binding the hydrophobic surfaces of the two microfibrils together (Cosgrove, 2018b). These may be the selective sites of CW loosening by expansins or by CXEG-type enzymes where the microfibrils slide or separate, perhaps at a rate that is influenced by the bulk viscoelasticity of the microfibril–matrix network (Park & Cosgrove, 2015). Disruption of such non-covalent bonds allows 'slippage' of carbohydrate polymers at load-bearing elements of the CW. Although the CW models assume non-covalent bonding between cellulose and hemicelluloses such as xyloglucan, *Equisetum* hetero-trans-$\beta$-glucanase (HTG) covalently attaches cellulose onto xyloglucan oligosaccharides in vitro. Interestingly, recombinant bacterial expansin EXLX1 strongly augmented the cellulose:xyloglucan endotransglucosylase activity that produces cellulose–xyloglucan covalent bonds in the CWs of structural plant tissues in vitro (Herburger et al., 2020).

The current view of the primary CW is represented by a mesoscale coarse-grained molecular dynamics model (Zhang et al., 2021a). The assembled epidermal CW is based on the supramolecular structure of cellulose and matrix polysaccharides that resembles (real) physics and tensile mechanics. The multi-layered CW has a cross-lamellate organisation in which individual layers (lamellae) of stiff cellulose microfibrils form a laterally interconnected network binding noncovalently to hemicellulose that is embedded in pectin, forming a gel-like matrix. Individual lamellar microfibrils are aligned in the same direction and appear anisotropic in terms of in-plane stress resistance; however, the complete (real) CWs, consisting of many lamellae (approx. 100) are highly isotropic. Interestingly, the simple non-covalent-bonding generated cellulose network in which fibril–fibril sliding of aligned cellulose bundles bears most of the stress despite frequent xyloglucan bridging between microfibrils, and pectin abundance. Overall, in this dynamic load-bearing network, tensile forces are transmitted primarily through direct lateral contacts between cellulose microfibrils, rather than by matrix polysaccharides. Thus, although the action of expansins and other wall-modifying proteins was not part of it, the model clearly highlights the importance of the lateral cellulose microfibrils contacts and its potential modulators (particularly expansins) in the overall transmission of in-plane tensile forces.

### 5.2. Expansin-mediated changes in the CW biomechanics

The CW can undergo several types of deformation that can be measured either in situ (ideally in living plant tissues) or in simplified models, most frequently using onion epidermis peels clamped in a custom-made mechanical testing device (Cosgrove, 1989; 2011; Durachko & Cosgrove, 2009; Durachko et al., 2017; Wang et al., 2020; Zhang & Cosgrove, 2017; Zhang et al., 2019a). In some cases, slightly more complex systems such as de-frosted *Arabidopsis* petioles (Park & Cosgrove, 2012a; Xin et al., 2020), cucumber and *Arabidopsis* hypocotyls (Boron et al., 2015; Cosgrove, 1989; Marga et al., 2005; Park & Cosgrove, 2012b) or wheat coleoptiles (Hepler & Cosgrove, 2019) have been used. The advantage of using onion epidermal peels is that the mechanical properties of isolated CW fragments can be measured, largely neglecting the contribution of neighbouring cells, cell size or shape that might possibly influence the results when using indentation-based (AFM) measurements

(Cosgrove, 2018b and references therein). However, new technologies such as non-contact, optical Brillouin spectroscopy are emerging as tools to probe biomechanical properties of CWs in developing organs at the cellular (Scarcelli et al., 2015) or tissue level (Elsayad et al., 2016; Samalova et al., 2020).

When CWs become mechanically softer/more pliant (meaning more easily deformed by out-of-plane mechanical force, see the Glossary), they do not necessarily result in wall relaxation and cell growth. On the other hand, $\alpha$-expansins cause in-plane stress relaxation and prolonged enlargement of CWs, but they do not change the CW viscoelastic properties, as measured by tensile tests (Cosgrove, 2018a; Yuan et al., 2001). In other words, reducing the wall stiffness does not necessarily lead to CW loosening. One such observation was made by Wang et al. (2020) with pectin methylesterase (PME) that selectively softened the onion epidermal wall yet reduced expansin-mediated creep. Similarly, driselase, a potent cocktail of wall-degrading enzymes, removed cellulose microfibrils in superficial lamellae sequentially, and softened the wall (reduced its indentation-measured mechanical stiffness), yet did not induce wall loosening (Zhang et al., 2019a).

In contrast to this, expansins, despite possessing no obvious enzymatic activity, are able to induce irreversible time-dependent expansion of CWs without affecting its compliance as discussed above. Expansins cause almost immediate in vitro CW extension, allowing to extend the cell length 100 times when compared to its meristematic initials (Cosgrove, 2016b and references therein). Thus, to loosen CW, expansins probably modify non-covalent bonds in the cellulose microfibril network, laterally interconnected with a possible contribution of xyloglucans bound to the hydrophobic face of the cellulose microfibrils (Cosgrove, 2018b and references therein). The consequent fibril–fibril sliding seems to allow CW extension and in-plane stress release of the multi-lamellate CW structure (Zhang et al., 2019a; 2021a).

## 6. Involvement of expansins in various aspects of plant growth and development

### 6.1. Cell elongation: The more (expansin) the better?

Expansins were identified as factors that primarily enhance cell elongation. The CW fraction from the actively growing (apical) portion of cucumber hypocotyls was able to induce creep of heat-inactivated cucumber hypocotyls when measured by a constant load extensometer. The observed CW extension required acidic pH and was also seen upon application of cucumber extracts to CW isolated from actively growing tissues (hypocotyls, leaves, petioles and coleoptiles) from other plant species. The CW extracts from the basal (non-growing) hypocotyls were unable to induce cell extension of apical hypocotyl fragments. Nonetheless, even the (active) CW extracts from the apical regions were unable to induce CW extension of the basal hypocotyl fragments, suggesting maturation-associated changes in CW structure limiting susceptibility to these extension-inducing factors (McQueen-Mason et al., 1992).

Cell expansion is a developmental response that is most frequently associated with upregulation of endogenous expansins in various tissues from a number of species. These include petiole elongation associated with *RpEXPA1* upregulation and CW acidification in response to ethylene entrapment following flooding in *Rumex palustris* (Vreeburg et al., 2005), enlargement of floral organs and internodes due to overexpression of *PhEXPA1* in petunia (Zenoni et al., 2011), changes in petiole and leaf-blade size associated with up- and down-regulation of *AtEXPA10* in

*Arabidopsis*, root hair-specific expression of *AtEXP7* and *AtEXP18* (Cho & Cosgrove, 2002), and *AtEXPA1*-mediated cell elongation in the *Arabidopsis* root transition zone (Pacifici et al., 2018).

However, the correlation between cell extension and expansin activity is not absolute. Only a partial correlation between the activities of LeEXP2 and LeEXP18 and cell elongation has been observed in tomato. This implies the existence of another factor, acting in concert with expansins, that may control growth under certain physiological conditions (Caderas et al., 2000). In line with that, chemically regulated expression of *CsEXP1* in tobacco suggested the existence of a specific developmental phase, when the leaf is sensitive to upregulated expansin (Sloan et al., 2009). Consistent with this, downregulating several expansins being transcriptionally active during the phase of maximal leaf-cell expansion (*AtEXPA1,3, 5* and *10*) using inducible amiRNA resulted in leaf growth repression in the latter stages of leaf development. Surprisingly, the smaller leaves had larger cells, suggesting organ and cell context-specific outputs of expansin gene expression (Goh et al., 2012). In rice seedlings with inducible *OsEXP4* expression, OsEXP4 protein levels were correlated with growth, but constitutive expression of the same gene resulted in growth retardation (Choi et al., 2003). Dose-dependent effects were observed in *Arabidopsis* (over)expressing cucumber *CsEXPA1* using a DEX-inducible system (Craft et al., 2005). While low levels of *CsEXPA1* were able to broaden leaf lamina, high levels had strong negative effects, particularly on the enlargement of fast-growing (expanding) tissues like hypocotyls or petioles (Goh et al., 2014). Finally, both overexpression of *CsEXPA1* and amiRNA-based downregulation of endogenous expansins (At*EXPA1,3, 5* and *10*) impaired hypocotyl elongation in etiolated *Arabidopsis* seedlings (Ilias et al., 2019). Overall, the action of expansins on CW enlargement seems to be specific, with regard to both dose (expression level) and the particular developmental context.

### 6.2. Do expansins control CW enlargement by modulating CW remodelling?

As with the examples in the previous sections, transgenic tomato lines with high levels of CsEXPA1 showed overall growth inhibition. Notably, hypocotyls from CsEXPA1 OE tomatoes were less sensitive to exogenously applied expansin in the constant-load extensometer assay (Rochange et al., 2001). The authors proposed that the observed CW tension resistance can be partly due to CW adaptation to the excessive amount of CW-loosening expansins through 'a decrease in the abundance or activity of secondary loosening agents, or stiffening of the CWs via other components (such as the de-esterification of pectins or extensin crosslinking)' (quote taken from Rochange et al., 2001).

There are several other pieces of evidence supporting a possible role for expansins as modulators of CW remodelling. Downregulation of *PhEXPA1* in petunia led to CW thickening and reduction in crystalline cellulose content, suggesting involvement of PhEXPA1 in the cellulose synthesis or deposition (Zenoni et al., 2004). Further in *PhEXPA1* OE CWs, the relative abundance of CW polymers was altered (in this case less pectin and hemicellulose, but unchanged cellulose content). Another example is overexpression of root-specific *OsEXPA8* in rice, leading to changed root architecture (longer main root, more lateral roots and root hairs), taller plants and larger leaves. The *OsEXPA8* overexpression was associated with lower (AFM-measured) CW stiffness and an increase in the polysaccharide/lignin ratio as measured using FTIR (Ma et al., 2013). The observed changes in the CW composition

could be achieved by changes in substrate availability due to the binding of expansins also to other CW polymers besides cellulose (Zenoni et al., 2004). In support of this mechanism, the CBM of strawberry expansin 2 (CBM-FaExp2) was shown to bind not only cellulose/xyloglucans but also other CW polymers including xylan and pectin. The presence of CBM-FaExp2 decreased the activity of CW degrading enzymes such as polygalacturonase, endoglucanase, pectinase and xylanase in an in vitro assay, probably due to CBM-FaExp2 binding to the enzyme substrates (Nardi et al., 2013). Notably, the CBM of FaEXP2 shows a high level of similarity to CBMs of AtEXPA1, AtEXPA2 and potato CBM-Pot-BG097738. Furthermore, the aromatic residues of CBM-FaExp2 are conserved in CBM-Pot-BG097738, and they were proposed to be involved in binding CW polysaccharides (Nardi et al., 2013). Thus, the CW stiffening recently observed in *Arabidopsis* lines with high levels of *AtEXPA1* (Samalova et al., 2020) could be explained by a similar mechanism, that is, interference of AtEXPA1 binding to CW components with enzyme activity mediating CW softening. Furthermore, expansin-mediated changes in the accessibility of CW-modifying enzymes were also proposed to be how EXP1-controlled fruit softening in tomato (Brummell et al., 1999). However, the role of feedback regulations leading to changes in the expression of genes for several CW remodelling proteins could also contribute to the *EXPA* overexpression-induced changes in CW composition (Ilias et al., 2019).

The role of the *C*-terminal CBM and its possible functional importance in recognising cellulose and/or other CW sugar polymers was highlighted by the work of Boron et al. (2015). The overexpression of *AtEXLA2*, a member of the expansin-like A family in *Arabidopsis* led only to a weak enlargement of etiolated hypocotyls. That was accompanied by CW thickening and decreased CW strength manifesting as higher rupture frequency (twice that of WT) under load during the creep test with a constant-load extensiometer. As AtEXLA2 is lacking the three conserved residues necessary for the CW loosening activity of the *N*-terminal D1 domain, the authors hypothesise a possible role for the *C*-terminal CBM in cellulose crystallisation and/or its affecting xyloglucan/cellulose interaction, leading to the observed defects in CW biomechanical properties. However, expansins may control CW remodelling independently of competition with CW modulating enzymes by binding to a wide spectrum of CW polymers as demonstrated for GbEXPATR in cotton. GbEXPATR represents a truncated version of its homologue GbEXPA2, lacking the *C*-terminal CBM. Interestingly, while the OE of *GbEXPA2* had no significant effects on the length of mature fibres, overproduction of GbEXPATR led to longer, finer and stronger cotton fibres, probably via a GbEXPATR-mediated delay in the onset of secondary CW formation (Li et al., 2016).

The CW acts as a sensing platform and plants use a dedicated system to control and maintain CW homeostasis that allows them to adapt to developmental changes as well as to environmental stresses. The wall composition and mechanical integrity are monitored by cell wall integrity (CWI) sensors and mechanosensitive ion channels (Hamann, 2015; Novakovic et al., 2018). CWI signalling involves the perception of mechanical and physical changes of the plant cell environment and the generation of signals that are amplified through feedback processes. Disruption of CWI results in activation of stress responses and CW modifications that might prevent the cells from further damage, including oxidative crosslinking, productions of ROS, jasmonic acid (JA), salicylic acid (SA), ethylene, lignin or callose depositions, alterations in pectin methylesterification and finally swollen root cells and root growth arrest caused

by the inhibition of cellulose synthesis (Gigli-Bisceglia et al., 2020; Van der Does et al., 2017). Interestingly, one of the proposed CWI sensors (reviewed in Rui & Dinneny, 2020), the GPI-anchored COBRA (COB) localises predominantly to longitudinal CWs and controls the orientation of *Arabidopsis* root cell expansion (Schindelman et al., 2001). COB was shown to be involved in the regulation of cellulose crystallinity and microfibril orientation (Roudier et al., 2005; Schindelman et al., 2001). Considering cellulose/CW matrix interaction as the primary target of EXPAs and the aforementioned role of PhEXPA in the control of cellulose crystallinity, the role of CWI and downstream feedback regulations in mediating the CW remodelling in a response to EXPA-induced changes in CW biomechanical properties cannot be excluded. However, the molecular mechanisms perceiving mechanical forces at the CW–plasma membrane interphase and controlling CWI-initiated adaptive responses remain largely unknown as it is difficult to separate them from integrated hormonal and stress signalling (Vaahtera et al., 2019).

Taken together, apart from their role in CW loosening, expansins seem to be involved in controlling CW properties and composition by interfering with the action of CW remodelling enzymes, possibly via mechanisms that are both dependent and independent of expansin interaction with CW carbohydrates (Figure 1b).

### 6.3. Organ primordia specification/outgrowth

Besides their role in organ growth, expansins were shown to be involved in the initiation of new organs both in the shoot and in the root. Sephacryl beads coated with expansin purified from cucumber hypocotyls disturbed phyllotaxis by inducing new leaf primordia on the shoot apical meristem (SAM) in tomato (Fleming et al., 1997). Endogenous *LeREXP18* was shown to be expressed in new leaf primordia in tomato (Reinhardt et al., 1998). Accordingly, local microinduction of cucumber expansin *CsEXP1* in the tobacco SAM was able to induce new leaf formation and reverse the direction of new primordia appearance. Furthermore, the induction of *CsEXP1* at the leaf margin changed the leaf shape by inducing ectopic leaf lamina formation (Pien et al., 2001). More recently, a possible molecular mechanism underlying the expansin-mediated primordia induction has been elucidated by placing expansin-controlled CW loosening into a previously described framework comprising a feedback loop between CW tension and microtubule orientation in the SAM (Armezzani et al., 2018; Hamant et al., 2008; Sassi et al., 2014). Briefly, mechanical stress in the complex tissue of growing SAM affects the microtubule cytoskeleton, and that in turn controls morphogenesis (Hamant et al., 2008). In parallel, auxin affects the cortical microtubule dynamics thus enhancing microtubule isotropy; together with auxin-induced softening of the CW, this seems to be sufficient to induce new organ outgrowth (Sassi et al., 2014). However, the changes in microtubule organisation were shown to activate the transcription of genes which potentially can induce CW loosening (*PME3, XTH9* and *EXPA15*) independently of auxin accumulation and transport. Conversely, interfering with wall loosening promotes changes in microtubule organisation (Armezzani et al., 2018).

In the root, cytokinin-induced *AtEXPA1* and CW acidification were suggested to induce the elongation and differentiation of cells leaving the root apical meristem (RAM) in the root transition zone (Pacifici et al., 2018), and this is somewhat analogous to new organ primordia in the SAM. However, more recent studies seem to confirm neither cytokinin-inducible *AtEXPA1* in the

root transition zone nor the role of *AtEXPA1* in controlling root growth (Ramakrishna et al., 2019; Samalova et al., 2020). Instead, *AtEXPA1* seems to be involved in radial swelling of the lateral root founder cell as an important determinant of asymmetric cell division, initiating the process of lateral root (primordia) formation (Ramakrishna et al., 2019). Interestingly, also here the asymmetric swelling of the lateral root founder cell is dependent on auxin signalling and position-specific reorientation of cortical microtubules (isotropic in the position of asymmetric swelling; Vilches Barro et al., 2019). This result is another puzzle in the emerging role of mechanical interactions between pericycle and endodermis cells in lateral root formation (Vermeer et al., 2014) and more generally the role of cytoskeleton dynamics in the determination of primary CW biomechanics and cell division (reviewed in Chebli et al., 2021; Robinson, 2021).

## 7. Expansins under abiotic stress

The transcripts of many $\alpha$-expansins are up-regulated under abiotic stress (Marowa et al., 2016; Tenhaken, 2015). Accordingly, genetic approaches have shown that enhanced expansin expression might contribute to stress tolerance to drought (Chen et al., 2016; Hao et al., 2017; Liu et al., 2019; Narayan et al., 2019; Yang et al., 2020), high salinity (Chen et al., 2017; 2018a; Hao et al., 2017; Lu et al., 2013; Yan et al., 2014; Zhang et al., 2019b), heat (Xu et al., 2007; 2014), cold (Peng et al., 2019; Zhang et al., 2018a), oxidative (Chen et al., 2018b) and heavy metal (cadmium) stress (Ren et al., 2018; Zhang et al., 2018b). Moreover, Han et al. (2012; 2015) described that overexpression of β-expansin *TaEXPB23* also enhanced tolerance to oxidative and salt stress, similar to the β-expansins *ZmEXPB6* and *ZmEXPB8* studied by Geilfus et al. (2015) and Wu et al. (2001) respectively. The changes in $\alpha$-expansin gene activity under various abiotic stresses in different plants are summarised in Table 1.

The mechanism of expansin action in mediating stress resistance is still rather unclear. Investigating CW biomechanics under abiotic stresses is often challenging, so the focus has predominantly remained at the molecular level on genes involved in CW remodelling and on transcriptional and proteomic changes. Concerning changes in the composition and structure of CWs, loss of water can cause enhanced bonding among individual wall components which can impact the biosynthesis and deposition of newly formed CW polymers. This can be seen, for example, during salt stress, when sodium ions might influence pectin cross-links and disrupt microtubule stability, which consequently influence cellulose deposition (Wang et al., 2016b).

Reactive oxygen species (ROS) and peroxidases may also play an important role in the process of CW remodelling. ROS production occurs under many different stress conditions, but it is also necessary for normal growth and development (Mittler, 2017) hence their production and quenching must be tightly controlled (Castro et al., 2021 and references therein). ROS are responsible for the initial cross-linking of phenolic compounds and CWs glycoproteins resulting in stiffening. On the other hand, wall polysaccharides might be directly cleaved by hydroxyl radicals and weaken plant CWs (Fry, 1998; Müller et al., 2009; Schopfer, 2001; Schweikert et al., 2000). Tenhaken (2015) proposed a simplified model in which he suggests that plant organ growth under stress is a conflict between the two processes. According to this model, growth arrest under abiotic stresses is possibly caused by ROS- and peroxidase-induced cross-linking of glycoproteins and phenolics esterified

with hemicellulose polymers, resulting in a dense network in which expansins and XTH do not have access to the xyloglucan substrate. If ROS production (stress) continues and all peroxidase substrates are depleted, ROS accumulation might lead to the formation of hydroxyl radicals, inducing the opposite effect, that is, cleavage of polymer chains. This results in CW weakening that enables further growth, comparable to growth under non-stress conditions. However, the experimental evidence for the model (Figure 1c) remains to be provided.

Interestingly, the action of expansins may result in enhancing the activity of CW-bound peroxidases in order to mitigate oxidative stress; however, the mechanism remains unknown (Han et al., 2015). The increased activity of covalently bound CW peroxidases was observed in transgenic plants over-expressing *TaEXPB23* and *Arabidopsis expb2* mutant showed a reduction in the activity and a decrease of oxidative stress tolerance (Han et al., 2015). Furthermore, expansin-mediated heat stress tolerance also seems to involve increased antioxidative capacity, photosynthesis rate and reduction of structural damage (Xu et al., 2014).

According to Wu et al. (1996; 2001)), root cell elongation is maintained at low water potential following enhanced expansin expression that enables plants to withstand drought conditions. This adaptive response, enabling roots to continue growing despite reduced turgor pressure, increases the root: shoot ratio allowing roots to explore the soil for water while limiting the water loss through leaves (Cosgrove, 2021). Furthermore, expansins were also proposed to be involved in increasing CW flexibility during the de- and rehydration processes in the resurrection plant *Craterostigma plantagineum* (Jones & McQueen-Mason, 2004).

## 8. Conclusions and future outlines

In contrast to the long-standing perception that considered the CW a rather static structure, passively delimiting the plant cell shape and providing mechanical support to plant bodies, the CW is a complex and highly dynamic structure, whose biomechanical properties have key consequences for a number of responses. Expansins are among the factors that allow plants to selectively change CW biomechanics, thus controlling plant growth and morphogenesis. As it is clear from our brief overview of the rich literature on the topic, there are several aspects of expansin action that are worth emphasising.

First, expansins seem to act in a manner that is dependent on both their dose and the particular developmental context. Second, CW sensitivity to expansin action seems to be actively controlled during the plant life cycle and in a location-specific fashion, and this is mediated by other factors including apoplastic pH. Third, expansins seem to control CW biomechanical properties not only by inducing creep but also by influencing CW remodelling, possibly through the modulation of substrate availability to other CW remodelling factors and/or CWI signalling. These effects might have important but different consequences for the downstream developmental regulations. It is therefore obvious that in order to comprehend the importance of expansin-regulated plant development and abiotic stress responses we will need a detailed understanding of the spatiotemporal specificity of expansin expression and its localization in living plant tissues. The existence of feedback regulatory loops between expansin activity/levels and expansin-modulated CW biomechanics might explain the dose-dependent and sometimes contradictory expansin effects. Moreover, functional redundancy among members of the expansin family is highly

**Table 1.** Overview of published evidence on expansin role in abiotic stress response.

| Gene | Plant species | Abiotic stress | Change in gene activity and/or the stress response | References |
|---|---|---|---|---|
| *TaEXPA2* | *Nicotiana tabacum* | Drought | *Triticum aestivum EXPA2* OE enhanced tolerance, increase in seed production | Chen et al. (2016) |
| *TaEXPA2* | *Triticum aestivum* | Drought | OE enhanced tolerance, RNAi—increased sensitivity | Yang et al. (2020) |
| 29 *EXPA*, 9 *EXLA*, 2 *EXPB* | *Camellia sinensis* | Drought | The expression levels of 16 expansins were high | Bordoloi et al. (2021) |
| *CplEXP1, CplEXP2, CplEXP3* | *Craterostigma plantagineum* | Drought | Increase in transcript levels of *Craterostigma plantagineum EXP1* | Jones and McQueen-Mason (2004) |
| *EaEXPA1, SoEXPA1, ShEXPA1* | *Erianthus arundinaceus* | Drought | High expression of *Erianthus arundinaceus EXPA1* | Narayan et al. (2019) |
| *EaEXPA1* | *Saccharum* spp. *hybrid* | Drought | OE enhanced tolerance | Narayan et al. (2021) |
| *ZmEXP1, ZmEXP5* | *Zea mays* | Drought | Increased transcript levels | Wu et al. (2001) |
| *NtEXPA4* | *Nicotiana tabacum* | Drought, salt | OE enhanced tolerance, RNAi—increased sensitivity | Chen et al., (2018a) |
| *NtEXPA11* | *Nicotiana tabacum* | Drought, salt | OE enhanced tolerance | Marowa et al. (2020) |
| *RhEXPA4* | *Arabidopsis thaliana* | Drought, salt | *Rosa hybrida EXPA4* OE enhanced tolerance | Lu et al. (2013) |
| *AnEXPA1, AnEXPA2* | *Arabidopsis thaliana* | Drought, cold | *Ammopiptanthus nanus EXPAs* OE enhanced tolerance | Liu et al. (2019) |
| *AstEXPA1* | *Nicotiana tabacum* | Drought, salt, heat, cold | *Agrostis stolonifera EXPA1* OE enhanced tolerance | Hao et al. (2017) |
| *PttEXPA8* | *Nicotiana tabacum* | Drought, salt, heat, cold, cadmium | *Populus tomentosa EXPA8* OE enhanced tolerance | Zhang et al. (2019b) |
| *TaEXPA2* | *Nicotiana tabacum* | Salt | *Triticum aestivum EXPA2* OE enhanced tolerance | Chen et al. (2017) |
| *AtEXP2* | *Arabidopsis thaliana* | Salt, osmotic stress | *exp2* increased sensitivity, *Arabidopsis thaliana EXP2* OE enhanced tolerance | Yan et al. (2014) |
| *AsEXP1* | *Agrostis scabra, Agrostis stolonifera* | Heat | The expression level of *Agrostis scabra EXP1* was highly upregulated in shoots | Xu et al. (2007) |
| *PpEXP1* | *Nicotiana tabacum* | Heat | *Poa pratensis EXP1* OE enhanced tolerance | Xu et al. (2014) |
| *TaEXPAs* | *Triticum aestivum* | Cold | Differential expression could be related to low-temperature tolerance or sensitivity | Zhang et al. (2018a) |
| *TaEXPA8* | *Arabidopsis thaliana* | Cold, drought | TaEXPA8 genes were induced by low-temperature and drought TaEXPA8 genes were induced by low-temperature and drought *Triticum aestivum EXPA8* OE enhanced low-temperature tolerance | Peng et al. (2019) |
| *PtoEXPA12* | *Nicotiana tabacum* | Cadmium | *Populus tomentosa EXPA12* OE enhanced Cd uptake and led to Cd toxicity | Zhang, et al. (2018b) |
| *TaEXPA2* | *Nicotiana tabacum* | Cadmium | *Triticum aestivum EXPA2* OE enhanced tolerance | Ren et al. (2018) |
| *TaEXPA2 AtEXPA2* | *Triticum aestivum, Arabidopsis thaliana* | $H_2O_2$ (oxidative stress) | The expression level of *TaEXPA2 was* upregulated, *Triticum aestivum EXPA2 OE* enhanced tolerance | Chen et al. (2018b) |

*Note*: TaEXPA8 genes were induced by low-temperature and drought.

likely, and this may require phenotype assays of multiple mutants in expansin genes. Further, understanding the expansin structure (either using experimental or structure prediction algorithms, see Figure 1 and the text above) and binding specificity will be necessary to elucidate the possible importance of expansins in regulating CW composition by interfering with CW remodelling factors. However, it should be emphasised that most of the experimental evidence on the possible role of expansins in CW remodelling originates from overexpression studies. Thus, more detailed studies employing, for example, cell type-specific endogenous promoters will be necessary to assess the possible role of expansins in CW remodelling.

Finally, developing tools allowing in vivo assays of quantifiable CW biomechanical properties at (sub)cellular resolution will be critical. Approaches combining biology, physics and mathematical modelling are particularly salient in order to integrate the vast array of complex observations that is expected from state-of-the-art visualisation methods, molecular biology/biochemistry and genetics studies.

## Glossary of used biomechanical terms

| | |
|---|---|
| Extensibility | In general, the ability of a material to be deformed by a tensile force. Wall extensibility is the ability of the CW to increase in surface area irreversibly during growth |
| Cell wall stress | A force exerted on the CW divided by the wall cross-sectional area perpendicular to the force application vector |
| Cell wall stress relaxation | A reduction in wall stress due to rearrangement of the load-bearing polymers in the cell wall |
| Cell wall remodelling | Chemical modification of CW components in which linkages between cell wall polysaccharides must be undone and reformed |
| Cell wall loosening | A molecular process causing wall stress relaxation (Cosgrove, 2018a). In other words, CW loosening is the sum of biochemical changes underlying the physical process of wall stress relaxation by creep |
| Wall creep | An irreversible, time-dependent CW deformation leading to modification of non-covalent bonds between CW polymers and allowing the fibril–fibril sliding |
| Cell wall softening | CW modification that makes the wall more deformable to out-of-plane mechanical force measurable by, for example, indentation techniques |
| Cell wall weakening | A process that reduces the force or energy needed to break the wall |
| Cell wall stiffening | A molecular process resulting in an increase in CW stress resistance. Stiffening can decrease the cell expansion rate or halt expansion under a given turgor |

Modified from: Chebli and Geitmann (2017), Cosgrove (1993; 2018a) and Zhang et al. (2019a).

## Acknowledgements

We are grateful to Prof. Olivier Hamant for his kind invitation and for the opportunity to provide our view on this highly interesting and dynamically developing topic. We thank the anonymous reviewers for their constructive and helpful comments.

**Financial support.** This work was supported by the Ministry of Education, Youth and Sports of CR from the European Regional Development Fund Project 'Centre for Experimental Plant Biology': No. CZ.02.1.01/0.0/0.0/16_019/0000738, LTAUSA18161 and the Czech Science Foundation (19-24753S and 22-17501S).

**Conflict of interest.** The authors declare no conflict of interest.

**Authorship contributions.** M.S., E.G. and J.H. performed the literature search, conceived the review structure and wrote the manuscript. J.H. drew Figure 1.

**Data availability statement.** All the data discussed in the review were obtained from the referenced papers. The AtEXPA1 (AT1G69530) structural prediction was downloaded from AlphaFold Protein Structure Database (https://alphafold.ebi.ac.uk/).

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
