## [Reviewer Report]

Dear Editor:

Herewith, please find the manuscript entitled "Expansin-mediated developmental and adaptive responses – a matter of cell wall biomechanics?" by Samalova et al., which we kindly ask you to consider for publication in the Quantitative Plant Biology journal. The review submission was invited by the Editor-in-Chief, prof. Olivier Hamant. 

The manuscript is new and not being considered for publication elsewhere.

In the manuscript, we provide an overview of the recent knowledge on expansin role in the various aspects of plant growth and development. Expansins were originally identified as factors controlling cell elongation via a process called cell wall (CW) loosening. In our brief overview we summarize our knowledge on expansins’ distribution in plant as well as non-plant species, their structure and mode of action. We describe the well-established concepts of the three decades of expansin research, but also highlight the alternative scenarios, including putative interaction of expansins with other factors mediating dose- and developmental-context specific regulation of CW enlargement as well as the possible role of expansins in the control of CW biomechanics and stress response by modulating CW remodeling. 

We believe that this type of review will be of interest for a broad audience within the scientific community and thus worthy of publication in the Quantitative Plant Biology.

We thank you for your time and consideration, 

Yours sincerely,

Jan Hejátko.

Ujezd u Brna, June 30, 2021

---

## [Reviewer Report]

*Comments to Author*: In the manuscript “Expansin-mediated developmental and adaptive responses – a matter of cell wall biomechanics?” M. Samalova and co-authors have provided a short review of expansin proteins: their structure, hypothetic mode of action and roles in plant growth, development and stress responses. I think this review will be informative and interesting for Quantitative Plant Biology readers, but several major and many minor points should be addressed before it can be accepted for publication.

Major points

1) Terminology related to plant biomechanical properties

I have the feeling that the authors do not always understand the meaning of some biomechanical terms they use. In some cases they denote biomechanical properties by terms that are not commonly accepted in this field. For example, the term “softer” is intuitively all right when used for the wall properties determined by microindentation techniques, but it is misleading for describing the wall mechanics determined by classical methods based on in plane wall deformations. As the readers might work in distant fields from plant biomechanics, I would suggest the authors to include a glossary of biomechanical terms used in the manuscript with their definitions. It would also be very useful to introduce the term “extensibility” in the beginning of the review. This term is crucial to the field of plant growth regulation at the cell wall level. The same refers to the terms “stress relaxation” and “cell wall loosening”. Many issues of terminology are considered in the classical review by Cosgrove (New Phytologist, 1993, 124:1-23). I will also offer some alternative terms in my detailed comments below.

2) The section “Methods used to investigates expansins” (lines 312-337) put in the end renders the whole review inconsistent. Only the first paragraph of this section (lines 313-321) deals with expansins, all the rest (lines 322-337) refers to new techniques to study cell walls in general. In my opinion, this part (lines 322-337) is a good subject for a separate review and should be excluded from the present manuscript. The first paragraph (lines 313-321) could be moved before the section “Cell wall biomechanical models”. Expansins exert their action only under acidic pH. Instead of the general description of wall-related techniques (lines 322-337), the manuscript would benefit from the inclusion of a paragraph on how the wall pH is controlled. Such data might explain the situations when expansin expression does not correlate with their action.

Detailed comments

L8-10. Please provide relevant references.

L11. The term “weaker” is better than “softer” here.

L12. Creep is TIME-DEPENDENT irreversible deformation.

L25. Please remove “see more detail in the text below”.

L28-29. It would be better to rephrase this sentence “the presence of these genes in… suggests…”

L32-33. “assist plant-microbe interactions in nature…”

L63. hydrolysing.

L64-65. It would be better to rephrase “…pH-dependent, expansin-mediated CW loosening promotes relaxation…”

L66-67. Please remove “see also later the text”.

L104. Be very careful with the term “soft” when referring to plant biomechanics. The problem is that only Cosgrove (2018 and later) decided to use the term “softer” both to the results of indentation tests (which is fine to me) and tensile tests (in this case “softer” is not commonly accepted usage). Consider the term “more pliant” to replace “softer”.

L106. In this case it would be better to rephrase “but they do not change the wall mechanics, as measured by tensile tests..”

L107. One way to rephrase is “Wall loosening does not always reduce the wall stiffness”. Alternatively, this idea can be expressed using different terms “more extensible cell walls are not necessarily more pliant cell walls, and vise versa”.

L108, 110. Here “softened” is fine, as the results come from AFM measurements.

L116-120. I would remove the phrase “as proposed by the multi-net growth hypothesis”. The point is that both FESEM and AFM reveal the most internal wall layer(s), while the classical multi-net growth hypothesis states that passive cellulose reorientations increase gradually toward outer wall layers (see Preston (1982) Planta 155:356-363).

L125-127. The results of stress-strain assays are very poor indicators of cell wall extensibility (Cosgrove 1993 cited above, and his recent works cited in your manuscript). Use “more pliant” instead of “more extensible” in L126.

L131. “is likely due”

L152. Please decipher what “EXLX1” refers to.

L159-160. I would suggest to rephrase “was able to induce creep…hypocotyls measured by a constant load extensometer. “Irreversible extension” and “creep” is not the same as discussed in Zhang et al (2019a) cited in the present manuscript.

L222, 225. “AtEXLA2”, please correct according to Boron et al. (2015).

L295-298. OH radicals have extremely high nonspecific activity in cleaving different wall polymers. As such they would rather induce CW weakening rather than CW loosening. So their effect must be very different compared with that of expansins (L106).

L302-303. Please specify how expansins may enhance the activity of CW-bound peroxidases.

L307-308. How expansins can maintain a higher cell turgor? What is the mechanism? Don’t you consider the option that expansins, due to their action on the wall mechanics, could maintain growth under lower turgor values (e.g. under draught and salinity stress)?

L315. Extensometer measures extension against time (as in the creep method). I would use “mechanical testing device” here.

L321. Please remove extra “(“.

L349. Please define CW remodeling somewhere in the beginning of your manuscript. I think it is better not to oppose creep and CW remodeling. Don’t you think that CW remodeling could be an integral part of the creep process?

Page 28 (Fig. 1 legend). Concerning “mimicking expansin-induced CW loosening”, please reconsider it keeping in mind my comment to L295-298.

---

## [Reviewer Report]

*Comments to Author*: The manuscript ‘Expansin-mediated developmental and adaptive responses – a matter of cell wall biomechanics?’ reviews the current knowledge on the expansins proteins in the context of the current cell wall models.

The expansins although being studied for a long time, have often proven to be tricky in the interpretation of their direct phenotypes. The effects are often subtle, probably very local, and specific function often masked due to redundancy. The link to a clear mechanism of action of these enigmatic proteins still remains a big question in the field. The review is timely in view of the increasing understanding of the ultrastructure and composition of the cell wall and techniques for measurement of biomechanical properties. The review would be of broad interest to the cell wall and biomechanics community among others.

The authors highlight these aspects in the review and extensively cover several recent advances in cell wall models and expansin in this context. The authors further discuss the role played by expansins in different developmental processes and in response to abiotic stresses. They detail approaches employed to study these proteins and highlight the challenges with unravelling the mechanism of expansin action with the current tools available.

Overall, this review is well very structured and written. Only a few comments to consider:

1. Overall, The link between pH and expansins is an important one that can be highlighted more in the text. Line 13 describes the loosening theory linking the turgor pressure and wall stress aspect well. The review could benefit from a more critical discussion on the link between auxin – pH and expansins in the context of acid growth theory and its fit with the current cell wall biomechanical models. Particularly to support the statement in Line 77 and 107. This might help delineate the role of expansin on cellulose, pectin and wall components further. Some useful references: Brummel et al., 1999; Arsuffi and Braybrook, J.Exp.Bot, 2018; Dunser and Kleine-Vehn, Curr. Op in Plant Bio., 2015, Fendrych et al., eLife 2016.

2. Line 154: The section could benefit from a few lines on the difference in the grass wall compared to eudicots and β-expansins in this context. Some useful references: Wang et al., Plant Phys., 2016; Valdivia et al., Sexual Plant Reproduction, 2009; Yennawar et al., PNAS., 2006)

3. Line 265: It would be interesting to extend the discussion initiated on the link between auxin – microtubules and SAM in line 246 in the context of lateral root initiation. A few recent works in this context by Vilches Barro et. al., Current Biology, 2020; Review: Robinson, New Phytologist, 2021; Chebil et al., Current Biology, 2021. The recurrence of expansin-mediated wall loosening associated with asymmetrically dividing cells, suggests an interesting link between expansins and i) their ability to respond to unique mechanical stress asymmetry sensed in these meristematic tissues; ii) potential for polar auxin flux to influence local expansion.

4. Line 308: Referen in the text to the commentary on overexpression of α-expansin in conference of drought tolerance in wheat – Cosgrove et al., 2021.

5. The Figure 1 has not been referenced in the main text. The scheme is rather minimal compared to the wealth of information covered in the review. The review would overall benefit if the figure could be expanded to present expansins in light of the current cell wall model and differences between cell wall loosening and remodelling covered in the review.

6. The abstract mentions that the review covers the role of expansins in stress response. The use of the terms “abiotic stress response” would be more appropriated here as it is the only stress covered in this review and will help distinguish from the term ‘stress’ in the biomechanical context.

7. Line 312: Additional references for methods and studies could be highlighted: Improved plant tissue friendly confocal Raman microscopy – combined view of wall chemistry and couple with AFM etc., for biomechanics: (Antreich et al., J.Exp.Bot., 2021, Gierlinger et al., 2012); Cellular force microscopy – Majda et al., Plant Cell Morphogenesis, 2019; Mechanoprobes – Michels et al., PNAS, 2020; single particle tracking (sptPALM)– Bayle et al., Nature Protocols., 2021; microfluidics – Yanagisawa et al., Plant & Cell Physiology., 2021.

Minor comments, some that might make it more accessible to non-cell wall specialists.

- As the terms such as wall creep, stress relation, wall loosening, wall softening, wall remodelling have been used extensively across the text, the readers could benefit from a Table or an Appendix defining these terms and linked into the manuscript early on.

- Line 25, The sentence ‘to the best of our knowledge’ would be better that ‘all’ plants species.

- Line 45: Would be worth including these references– cotton (Lv et al., BMC Plat Biology, 2020); brassica (Li et al., Plant Phys and Biochem, 2021).

- Line 113: The main message from the statement in line 113 is not very clear and could be simplified.

- Line 112-113: Please provide references on the model(s) the authors refer to in these lines.

- Line 133: Nuclear Magnetic Resonance and can abbreviated (NMR).

- Table 1 – Additional references for wheat and drought tolerance, Calderini et al., New Phytologist, 2020, sugarcane Narayan et al., 2021.

---

## [Reviewer Report]

*Comments to Author*: Dear Jan Hejatko,

Thank you for submitting your manuscript to Quantitative Plant Biology and please accept my sincerest apologies for the unusually long review process. We have now received comments by two expert reviewers and, as you can see below, they generally endorse your manuscript, but raise several important points that should be addressed in a revised version. In addition, I would also ask you to consider the following points:

-The manuscript covers a lot of topics and therefore in some instances can seem redundant with other published reviews. Sometimes, studies that reach opposing conclusions to the ones mentioned are neglected. Despite the wide variety of topics, biotic stress is not elaborated upon (although many expansins are encoded in the genomes of plant pathogens), while the “mechanics” part, even though featured in the title, could benefit from further attention, as mentioned by the reviewers.

-Please make sure to carefully and precisely differentiate between the properties, activities, and putative biological roles of the different classes of expansins. For example, the acid growth scenario does not apply to microbial or beta-expansins. Another example is the model of Venezuela-Riffo (line 74), which only applies to alpha-expansins. 

-Please make sure to distinguish between hypotheses/speculation and conclusion supported by data. Some hypotheses restated in the manuscript seem to be obsolete. 

-A lot of evidence suggests that the traditional cell wall model introduced at the beginning of the manuscript is outdated, which is relevant in the context of possible roles of expansins

-Please double check the section on industrial applications of expansins

-Lastly, a word of caution could be helpful with respect to the interpretation of phenotypes induced by ectopic expression of expansins in planta, as these could be invoked by secondary responses, which have not been studied in detail so far.

---

## [Reviewer Report]

*Comments to Author*: After reading a resubmitted version of the review article “Expansin-mediated developmental and adaptive responses – a matter of cell wall biomechanics?” by M. Samalova et al. I can see that the authors did a good job addressing the points raised by the reviewers and the editor, so I can recommend its acceptance for publication in Quantitative Plant Biology. Still I have some suggestions on how to further improve the manuscript.

1) In the section ‘Do expansins control CW enlargement by modulating cell wall remodeling’ the authors emphasize one mechanism by which expansins may exert this effect: their interference with the action of CW remodelling enzymes. While this particular mechanism could really be instrumental, I think it is worth mentioning a more general and possibly more important mechanism: cell wall integrity sensing and maintenance. Irrespective of the cell wall model proposed, cellulose microfibrils have always been considered as crucial components for its mechanics. Cellulose has also been implicated in the strong effects of expansins on the wall physical properties. Thus, it is tempting to speculate that large artificial changes in expansin levels (by overexpression, etc.) would induce some feedback signaling from the cell wall to the cell with the wall remodeling as a final result of this chain of events. Please briefly consider this option in your review. Here are some relevant reviews on the cell wall integrity sensing and maintenance:

Voxeur, Hofte (2016) Glycobiology, 26, 950–960

Wolf (2017) Biochem J. 474, 471–492

Vaahtera et al (2019) Nature Plants 5, 924–932

2) The manuscript benefited from the inclusion of a glossary of terms related to biomechanics (page 34). Please put the glossary in the beginning of the final version of your review. Try to use more concise wording for every term (ideally not more than one sentence) and avoid using alternative versions for it (establish your priorities). I also think that the definitions of some terms have to be clarified. You may disagree with my suggestions below, but please consider them.

- Extensibility. Its definition in the current version of the MS was adapted from Chebli & Geitmann (2017) and relates to the general meaning of this word in English. However the term ‘cell wall extensibility’ has one very important aspect not reflected by the word ‘extensibility’ in its general meaning. ‘Cell wall extensibility’ deals with the situation IN VIVO. A large number of known (and possibly many unknown) in vivo processes contributes to the wall extensibility: cell wall composition, its continuous modification by endogenous enzymatic and non-enzymatic proteins, incorporation of new structural components to the wall and the direction of their deposition, the level of wall hydration, apoplastic pH and ionic conditions, etc. All currently used biomechanical tests provide better or worse estimates of the wall extensibility, and some of their metrics are irrelevant to the extensibility and growth control. Keeping this in mind, I think that the old definition of the wall extensibility by Cosgrove (1993) in its slightly updated form (Cosgrove (2016) J. Exp. Bot. 67: 463-476) is the best one: “wall extensibility is the ability of the cell wall to increase in surface area irreversibly during growth”.

- Cell wall stress. I think it would be more accurate to use a slightly adapted definition from Chebli & Geitmann (2017). Cell wall stress is force exerted on the cell wall divided by the wall cross-sectional area perpendicular to the force application.

- Cell wall stress relaxation (with slight modification from Chebli & Geitmann (2017)) is a decay in the wall stress due to rearrangement of the load-bearing polymers in the cell wall.

- Cell wall remodeling. Your current definition is fine to me.

- Cell wall loosening is a molecular process causing wall stress relaxation (Cosgrove, 2018). In other words, cell wall loosening is biochemical changes underlying the physical process of wall stress relaxation.

- Wall creep is an irreversible, time-dependent CW deformation.

- Cell wall softening. Your current definition is fine to me.

- Cell wall weakening. Your current definition is fine to me.

- Cell wall stiffening is a molecular process resulting in an increase of CW stress resistance.

3) Lines 207-227. Although the article by Zhang et al. (2021a) is interesting and seminal, retelling this story on a half of page in your review is not the best option. Please summarize the main findings of Zhang et al. (2021a) in a much more concise way.

4) Please ask your colleagues, who are native speakers of English, to read the final version of your manuscript and correct some minor grammatical errors remaining.

Some minor corrections:

Lines 17, 78, 112, 121, 151, 159 and throughout the text. Please remove the phrases like ‘as discussed further in the text’, ‘see also later in the text’, ‘see also below’, etc. An interested reader will definitely reach the end of your article.

Line 112. Please do not emphasize ‘xyloglucan-interconnected cellulose microfibrils’ because this model of the main cell wall load-bearing network might not be valid, as you discuss further in the text.

Line 121. … by allowing ?

Line 180. brackets are not needed.

Line 182. due to

Line 255. This is overstated. Expansins cause almost immediate in vitro cell wall extension.

Lines 256, 259. therein

Line 267. when measured

Line 421. a reduction in the activity

Lines 444-445. Please supplement this sentence according to my suggestion 1)

Line 447. we will need

Line 446. to Prof.

Page 37 (legend of Fig. 1B) expansins might contribute to CW remodeling, too.

---

## [Reviewer Report]

*Comments to Author*: I appreciate the responses to the comments and the revised manuscript. The glossary in particular is a useful addition.

Some minor edits:

Line 79: The Alpha fold structure predictions are interesting but what is the advantage. A line on benefit in the context of expansin research?

Line 140: Suggest to rephrase as a more general reasons for non-transcriptional regulation.

Line 115 and Line 136 convey mixed messages on the sequence of events in the cell wall. I would suggest clarifying Line 136.

Line 167-169: Rephrase the statement to simplify the message.

Line 240: Please change to `non-contact`

Glossary: Provide example of an out-of-plane force (as has been done for in-plane). Would help put the force into cellular context.

Table 1. Please edit Column 1 on Gene names – the prefix of species lower case.

Figure 1. Suggest inclusion of a one line header for part B and C of figure in the legend. A version of something along the lines: Part B. Model of expansin domains and potential actions...

---

## [Reviewer Report]

*Comments to Author*: Dear Jan Hejatko,

We have now obtained the reviews for your revised manuscript and I am happy to tell you that your work is accepted for publication. As you can see, both reviewers greatly appreciate the changes you made; both also provide some additional suggestions that I would encourage you to consider. Thank you very much for submitting this review to QPB.

Best regards,

Sebastian

---

## [Reviewer Report]

*Comments to Author*: Dear Jan Hejatko,

thank you again for submitting your work to QPB and thank you for addressing the remaining points of the reviewers. I am happy to confirm that your review is now accepted for publication.

Best regards,

Sebastian Wolf